# Large-Scale Preparation of Uniform Millet Bread-like Durable Benzoxazine-Phthalonitrile Foam with Outstanding Mechanical and Thermal Properties

**DOI:** 10.3390/polym14245410

**Published:** 2022-12-10

**Authors:** Wenwu Lei, Dengyu Wang, Qi Liu, Kui Li, Ying Li, Fei Zhong, Qiancheng Liu, Pan Wang, Wei Feng, Xulin Yang

**Affiliations:** 1School of Mechanical Engineering, Chengdu University, Chengdu 610106, China; 2Chemical Synthesis and Pollution Control Key Laboratory of Sichuan Province, China West Normal University, Nanchong 637009, China; 3Sichuan Province Engineering Technology Research Center of Powder Metallurgy, Chengdu University, Chengdu 610106, China; 4Institute for Advanced Study, Chengdu University, Chengdu 610106, China

**Keywords:** foaming temperature, phthalonitrile foam, large-scale production, benzoxazine-phthalonitrile foam

## Abstract

It is essentially important to develop durable polymer foams for services in high-temperature conditions. The current study reported the preparations and properties of a high-performance benzoxazine-phthalonitrile (BZPN) foam by utilizing azodicarbonamide and tween-80 as the blowing agent and stabilizer, respectively. Rheological and curing studies indicated that the appropriate foaming temperature for BZPN foam is below 180 °C, and its foaming viscosity window is below 20 Pa·s. Guided by these results, uniform millet bread-like BZPN foams with decimeter leveling size were successfully realized, suggesting the high prospect of large-scale production. The structural, mechanical, and thermal properties of BZPN foams were then investigated in detail. BZPN foam involves a hierarchical fracture mechanism during the compressive test, and it shows a high compression strength of over 6 MPa. During a burning test over 380 °C, no visible smoke, softening, or droplet phenomena appeared and the macroscopic structure of BZPN foam was well maintained. Mechanically robust, flame-retardant, and uniform large-size BZPN foam are promising light durable materials with high service temperatures, i.e., as filling materials even in a very narrow pipette.

## 1. Introduction

Polymeric foam is a typical two-phase material consisting of gas dispersed in a solid polymer. It is generally produced by adding a physical and/or chemical blowing agent in the polymer host [1,2]. The chemical blowing agent generates gases during the chemical reaction [3], whereas the physical one introduces a reversible template/hollow component into the polymer host [4,5], or produces volatiles via the physical process [6]. Due to its two-phase structure, the polymer phase can guarantee mechanical strength while the gas phase can ensure exceptional lightness and low conductivity. Compared with bulky polymers, polymer foams exhibit many superior properties such as high specific strength and good thermal/sound insulation. They are now irreplaceable in various applications including construction, packaging, filtration, electronics, and aerospace materials [7,8,9,10]. For example, the latest COVID-19 eruption makes the porous polymer materials irreplaceable, especially in personal protective equipment including masks, respirators, and visors [11]. Presently, the polymer foam industry is mastered by polyurethane foam, which accounts for 50% of annual global production [12,13]. Others are mainly polyethylene [14], polystyrene [15], polyimide [16], epoxy [17], and phenolic foams [18]. These foams, based on universal polymers, have good mechanical properties, but their thermal resistances are limited and most of them are easily burned [19]. Herein, it is essentially important to develop durable polymer foams for services in the high-temperature-related harsh environment.

Phthalonitrile (PN) resins, as a special polymer to meet the material requirements of the US Navy Military Standard (MIL-STD-2031), are famous for their superior thermal, thermo-oxidative, and flame resistances [20,21,22,23]. In view of these unique properties, PN resins are promising precursors for porous materials. Recently, Hou et al. [24] reported graphene/PN aerogels with porous structure by a solvothermal and carbonation process for oxygen reduction. Zhang et al. [25] prepared carbon foams from PN resins for use in electromagnetic shielding. By the carbonation of PN/urea/KOH mixtures, Weng et al. [26] prepared N/O co-doped hierarchical porous carbon for supercapacitor applications. These advances confirmed that PN resins were ideal polymer matrices for advanced porous carbon composites. However, these porous materials are mainly in the form of powder, which is weak in macroscopic mechanical strength. The report on the PN-based bulky polymer foam is quite limited. On the other hand, our group recently found that azodicarbonamide (AC) can take double roles of curing agent and blowing agent in the benzoxazine-phthalonitrile (BZPN) systems [27]. The thermal stabilities of the obtained BZPN foam are outstanding, but its compressive strength is fragile and only limited in the ranges of 0.399 MPa to 1.793 MPa [27]. The mechanical properties of these BZPN foams need to be further improved to cater to long-term service at elevated temperature exposure.

Herein, the current study mainly deals with the large-scale preparations of BZPN foams with outstanding mechanical and thermal properties. BZPN is selected as a polymer matrix because it is bi-functional and instigates the advantages of PN and benzoxazine reins [27,28,29]. Azodicarbonamide (AC) is selected as a chemical foam agent because its decomposition temperature matches the curing temperature of BZPN [30,31]. Tween-80 (TW) is used as a foam stabilizer due to its intrinsic compatibility with BZPN. By using BZPN/AC/TW blend as the foam precursor and through the rheological and curing studies, key parameters for large-scale production of BZPN foams can be proposed. The effect of foaming temperature on the structural, mechanical, and thermal properties of BZPN foams was systematically investigated and assessed. This will contribute to the development of large-scale preparation and practical application of BZPN foams and provide a basis for future work on durable polymer foams.

## 2. Experimental Section

### 2.1. Materials

The gold bi-functional benzoxazine-phthalonitrile (BZPN) powder, with a density of 0.76 g/cm^3^ and a softening temperature of around 120 °C, was provided by Shunde Great New Materials Co., Ltd. (Foshan, China). Its chemical structure is also given in Appendix A. Yellow azodicarbonamide (AC, CAS#: 123-77-3) powder and viscous translucent tween-80 (TW, CAS#: 9005-65-6) liquid were given by Chengdu Kelong Chemical Co., Ltd. (Chengdu, China).

### 2.2. Preparation of BZPN Foam

Figure 1 illustrates the formation steps of BZPN foam. Typically, certain weights of TW-80 and deionized water were added to form a colorless transparent solution. Then, AC was introduced and stirred for 30 min, giving a light yellow dispersion. Afterward, BZPN was added and stirred for 30 min, resulting in an earthy yellow viscous mixture. The AC content is controlled with 2 wt% in BZPN system based on our previous study (a saturated foaming efficiency of around 2 wt% AC) [27]. The weight ratio of BZPN:AC:TW was designed as 100:2:10. Finally, the above mixture was transferred into a mold at 80 °C for 2 h to remove moisture and then foamed at a certain foaming temperature of 120 °C, 140 °C, 160 °C, and 180 °C, followed by a temperature/time curing procedure, as given in Table 1. Consequently, a series of black porous BZPN foams foamed at various temperatures were obtained.

### 2.3. Characterization

Dynamic rheological analysis (DRA) of BZPN foam precursor (BZPN/AC/TW powder) was characterized by AR-G2 in the air. BZPN/AC/TW powder (about 1.0 g) was melted between two parallel plates (diameter: 25 mm) at 1 Hz via two kinds of modes: (1) temperature scan conducted at different temperatures from 70 °C to 300 °C at a heating rate of 2.5 °C/min, and (2) time scan was conducted at a fixed temperature. When it reached the preset isothermal temperature, the chamber was opened and BZPN/AC/TW powder was loaded between the plates. The chamber was then closed and the data was collected once the equilibrium temperature returned to the preset temperature (about 3 min). Differential scanning calorimetric (DSC) scans of BZPN/AC/TW foam precursor from 30 °C to 300 °C were conducted via Q100 at 10 °C/min in nitrogen. Fourier transform infrared (FTIR) spectra from 4000 cm^−1^ to 400 cm^−1^ of BZPN foam precursor were conducted by a PerkinElmer Spectrum 100 spectrometer at the resolution of 2 cm^−1^ and 32-time scanning in air. Scanning electron microscopy (SEM) images of the final foams after gold coating were conducted on JEOL-4700F at 20 kV. The compressive performances of BZPN foams were accessed on CMT 5205 at 1 mm/min, and the results were averaged and reported from 5 samples. Flame test of BZPN foam was directly burned on an alcohol lamp. The weights before and after burning were measured to calculate the mass retention. The digital photos during the whole burning test were also recorded by an IMX 600 camera (F1.8) from HUAWEI P20 Pro cellphone. Thermogravimetry (TGA/DTG) of BZPN foam was carried out by Q50 between 50 °C and 800 °C at 20 °C/min in N_2_ with a purge of 40 mL/min.

## 3. Results and Discussions

### 3.1. Rheological Behavior

Polymer foaming usually involves bubble nucleation, growth, and stabilization. For a good foam formation, the key is the synchronization between the blowing agent decomposition and the polymer host viscosification [32], which are first decided by the processing temperature. It has been reported that the decomposition temperature of AC is from 130 °C to 200 °C [30,31], but the viscosification temperature of the BZPN/AC/TW system remains unknown. It is essentially important that the bubble nucleates and grows in a specific viscosity range (foaming viscosity window) [32]. Herein, the complex viscosity (η*) of BZPN foam precursor from 70 °C to 300 °C was first provided in Figure 2a. With the increasing temperature, the η* of BZPN/AC/TW foam precursor decreases first (<100 °C), fluctuates gently (100 °C–200 °C), increases abruptly (200 °C–240 °C), and finally, stabilizes (>240 °C), which matches the softening, melting, gelling, and curing of the BZPN/AC/TW foam precursor, respectively. The state of the BZPN/AC/TW foam precursor varied from solid to liquid, rubber, and solid, respectively.

According to classical rheological theory, the storage modulus (G′) implies the elastic property but the loss modulus (G″) reveals the viscous property of a melt system. G′ is a reflection of ‘stiffness’, and G″ is an indication of ‘softness’ [33,34]. Herein, G′ and G″ can provide the inner state information of the BZPN/AC/TW foam precursor and thus guide the BZPN foam formation. By comparing G′ with G″, the curve in Figure 2b could be classified into three regions:

G″ > G′> Gions c< 180 °C. The BZPN foam precursor is viscous and in a liquid state. In this region, bubble growth is easy, whereas bubble stabilization is relatively difficult. If the gas is released by a blowing agent in this region, it will quickly diffuse through the polymer and form a well-dispersed gas phase in the polymer melt phase. Meanwhile, if the viscosity of BZPN melt just enters the foaming window, the cavities will grow and be stabilized, and uniform foam can be expected. Otherwise, the cavities will collapse before stabilization, and no foam will be formed [35].

G″ = G′= Glization, °C–220 °C. The BZPN foam system is close to a rubber state. During this process, the viscosity of BZPN is relatively high and it is beneficial for bubble stabilization. However, bubble growth can be stopped by increasing viscosity. Once the gas escapes before the foaming viscosity window, bubble nucleation tends to be impossible. Alternatively, bubble growth is limited by the high elastic modulus that prevents the deformation of the polymer host [36].

G′ > G″> Gt [36]>220 °C. The BZPN foam precursor is in a solid-like state where bubble stabilization is impossible. On the other hand, this temperature is beyond the AC decomposition temperature (130–200 °C) [30,31]. Herein, in this region, AC in BZPN foam precursor has been totally decomposed, making bubble nucleation and growth impossible. That is, BZPN foam cannot be formed in this region.

From the above analysis, the viscosity foaming window temperature of BZPN foam precursor can be only in the temperature region of G″ and G′ (<180 °C) or in the very beginning of G″ = G′ (180 °C–220°C). To monitor the η* change of BZPN foam precursor in real-time, the η* curves of BZPN foam precursor as a function of time at various temperatures were conducted and given in Figure 2c,d. For comparison, Table 2 also gives the detailed η* values under these temperatures. At 120 °C, 140 °C, and 160 °C, little change of η* indicates that the curing reaction performs very slowly. It is easy for bubbles to disperse through the polymer matrix. Thus, a uniform dispersion of gas in the BZPN phase can be expected if the bubble stabilization simultaneously happens [32]. As can be seen in Table 2, η* at 120 °C fluctuates with very small deviations (<0.1 Pa·s, this value is close to the viscosity of Newtonian fluid such as fruit juice [37]), indicating that the viscosification of the BZPN was almost non-existent at this temperature. However, the η* at 140 °C and 160 °C increases steadily with increasing time, suggesting that viscosification of the BZPN does happen at these temperatures. Moreover, the η* values at 140 °C and 160 °C are below 20 Pa·s, close to the viscosity of non-Newtonian fluid like honey [38]. It is viscous but has flowability. The bubbles can diffuse, grow, and stabilize, and thus a uniform foaming structure can be expected. However, the η* at 180 °C is as high as 878.2 Pa·s, and it increases quickly with time. An abrupt η* increase to 2068 Pa·s was observed after about 30 min. This ultrahigh viscosity means that flowability is lost and gas dispersion is difficult. To conclude, the appropriate viscosity foaming window for BZPN/AC/TW foam precursor is below 20 Pa·s.

### 3.2. Curing Behavior

For thermosetting systems, the rheological elastic and viscous behavior, in excess, is dominated by its curing reaction process. If bubbling occurs first without successive curing, bubbles collapse. Herein, the curing behavior of the BZPN foam precursor was characterized by DSC in Figure 3a. Bi-functional BZPN shows two curing peaks at 239 °C and 267 °C, corresponding to the oxazine ring-opening and phthalonitrile ring-forming curing reactions [26,27], respectively. The temperatures of these curing peaks are lowered in BZPN/AC/TW foam precursor. These results indicated that the active hydrogen-containing blowing agent of AC and foam homogenizer of TW-80 could promote the curing process of bi-functional BZPN, in line with the findings of a previous study [27]. Nevertheless, the onset temperature of the oxazine ring-opening reaction is over 180 °C from the DSC curve, and it is higher than the viscosity-foaming window temperature from the above DRA findings. This does not mean that the oxazine ring-opening reaction (or viscosification of the polymer) was not in progress below 180 °C. The direct evidence was provided in Figure 3b, where the DSC curves of the samples heated at 120 °C, 140 °C, 160 °C, or 180 °C for 1 h were displayed.

It can be seen from Figure 3b that both the oxazine ring-opening peak and phthalonitrile ring-forming peak were reduced obviously as the foaming temperature increased. Especially, these two peaks overlapped in the BZPN/AC/TW foam precursor after heating at 180 °C for 1 h. On the one hand, these results revealed that the DRA is more sensitive than DSC in the characterization of the BZPN/AC/TW foam precursor. On the hand, these results directly suggest that the curing reaction of BZPN was being carried out below 180 °C, in line with the DRA observations in Figure 2c,d that the η* and G′ increases steadily over 140 °C due to the curing reaction of BZPN. This inference is further confirmed by the FTIR results, as shown in Figure 3c. As the foaming temperature increased from 120 ℃ to 180 ℃, the characteristic bands at 2234 cm^−1^ for phthalonitrile and 937 cm^−1^ for benzoxazine groups obliviously decreased. In BZPN/AC/TW systems, both AC and TW are active hydrogen-abundant compounds. Previous studies have reported that these compounds were the common curing agents for BZ and PN groups [20,21,22,27,28,29]. The active hydrogen interacts with benzoxazine groups at the carbon atom (O-C-N) in the oxazine ring, resulting in the secondary amine-like product [29]. Meanwhile, the interaction between active hydrogen and phthalonitrile groups follows the nucleophilic substitution mechanism at the carbon atom (Ar-C≡N) in the nitrile group, leading to heterocyclic polymers [28]. As can be observed in Figure 3c, the small absorption band at 3330 cm^−1^ was attributed to the N-H stretching vibration in secondary amine whereas peaks at 1718 cm^−1^ and 1480 cm^−1^, 1360 cm^−1^, and 1010 cm^−1^ corresponded to the formation of heterocyclic polymers including isoindoline, triazine, and phthalocyanines [22,27,28,29], respectively.

Combining the rheological results with curing studies, it can be concluded that the appropriate foaming temperature of BZPN foam precursor is no more than 180 °C, and its viscosity foaming window is below Pa·s. Guided by these results, the BZPN foam precursor was foamed at different temperatures. The effect of foaming temperature on the structures and properties of final foams is investigated.

### 3.3. Foam Structure

Figure 4 provides typical digital pictures of BZPN foams formed at 140 °C. As can be seen in Figure 4a,b, regardless of its color, BZPN foam looks like millet bread—a traditional Chinese snack. It is black, with a visibly uniform porous structure. Moreover, BZPN foam precursor can be foamed into any shape according to the mold shape, i.e., a cylinder shape (Figure 4c). It can be used as filler material even in a very narrow pipette (Figure 4d). A uniform porous structure with a centimeter leveling also suggests the high prospect of large-scale production for BZPN foams. Especially, it is also worth noting that BZPN is a class of thermosetting resin, and the current method can be used to prepare other types of BZPN foams and BZPN/filler composite foams with various properties, i.e., tri-functional allyl-containing BZPN foam, BZPN/CNT conductive foam, and BZPN/Fe3O4 magnetic foam. This will further widen the applications of these foams.

Figure 5 also provides the SEM images of BZPN foams. Overall, all samples showed a typical porous structure, once again confirming that the appropriate foaming temperature of BZPN foam precursor is below 180 °C. There are, however, some differences among the foam structures of these samples foamed at various temperatures. As can be observed in Figure 5a, the cellular structure of BZPN at 120 °C is spherical, including big pores surrounding various small cells. At low temperature, both the rate of AC decomposition and that of BZPN viscosification is quite slow. The bubble nucleates and grows, but some bubbles have not been stabilized yet. As a result, partial bubbles collapsed or escaped, leaving some visible voids in the matrix or small holes in the cell walls. For BZPN foam at 140 °C in Figure 5b, the cells tend to be uniform and regular. The aggregated cell structure suggested a good synchronization between bubble growth and bubble stabilization. From the DRA results in Table 2, the steady η* increase at 140 °C is helpful for bubble stabilization. As the foaming temperature increases to 160 °C, the viscosity also increases and bubble growth can be blocked by the higher viscosity. Consequently, in Figure 5c, the diameter of cells becomes smaller. When the foaming temperature is 180 °C, the initial η* is too high. Bubble development is rather difficult because the polymer matrix loses flowing and deformation ability. Herein, BZPN foam at 180 °C seems to lose its spherical structure with small holes in each cell (Figure 5d). It should be mentioned that the general statistics for cell distribution are not applicable due to the irregular cell shape (especially BZPN foam at 180 °C) and relatively large pore size (about 400 um). In addition, compared with BZPN/AC foams [27], BZPN/AC/TW foams showed a better-foamed structure, indicating that the usage of TW is helpful for foam formation due to its stability for bubbles [39].

### 3.4. Mechanical Property

The mechanical property of BZPN foam was characterized by the compression test, as provided in Figure 6a. The stress–strain curves were ‘noisy’ over the whole strain region. After initial failure, the compression stress slightly dropped and then increased. This phenomenon was repeated in every BZPN foam several times until the stress dropped obviously at the end. Such a phenomenon makes the accurate measurement of compression strength and modulus difficult. Nevertheless, it can be still observed from Figure 6a that the compression strength of BZPN foams formed at 120 °C, 140 °C, and 160 °C are all higher than 6 MPa, whereas foam formed at 180 °C showed the worst compression performance. Figure 6b further gives the compressive stress at a fixed strain of 10%. It can be seen that the compressive performance decreased as the foaming temperature increased from 120 °C to 180 °C. In comparison with the polymer/hollow glass microspheres syntactic foams [5,40], polymer foams prepared by the chemical foaming method [6,27,41,42] showed a smaller density and thus lower compressive performance. However, whether it adopts a chemical foaming method or physical foaming one, as previous studies described, the compressive properties of foaming materials usually decrease with the decreasing density [41,42]. For example, the strength of BZ foam decreased from 12.4 MPa to 5.2 MPa as its density decreased from 0.407 g/cm^3^ to 0.273 g/cm^3^ [41]. In the case of BZPN foam, as the foaming temperature increases from 120 °C to 180 °C, its density decreased from 0.46 g/cm^3^ to 0.40 g/cm^3^. Consequently, the compressive performance of BZPN foam correspondingly decreased. In addition, from Figure 5d, the high foaming temperature of 180 °C leads to an uneven foam structure, which is disadvantageous to stress transfer. Consequently, it showed the worst compression performance.

Figure 6c,d gives the SEM images of a BZPN foam formed at 140 °C after a compressive test. At the top and bottom in Figure 6c, a part of broken cells can be observed, leaving obvious fracture debris on the surface of the cells. In the middle of Figure 6c, some of the bubbles are deformed or compacted, suggesting that the bubbles are aggregated under the compression load. Moreover, the outline of bubbles in Figure 6c,d tends to be unclear in comparison with the original image in Figure 5b. These results indicate that there is a densification process in the BZPN foams during the compression test. Jalalian et al. also observed a densification phenomenon in the phenolic-based foams on the macro level during the compression test [6]. Based on these findings, the failure mechanism can be illustrated in Figure 6e. When the compression load was put on the BZPN foam, the stress accumulated quickly in the bottom or top layers of the BZPN foam. Due to the isolated pore structure and intrinsic brittleness of thermosetting resin, on the bottom or top plane, the bubble position is the mechanically weak location. Herein, the stress cannot be well transferred over the whole specimen through bubble buckling. The local concentrated stress immediately exceeded the limit of bubbles at the surface plane, making the bubble at this layer first. Consequently, the cracked bubbles were further compacted and they could enable stress transfer to the next layer resulting in the progressive fracture of the species until the failure of the main foam body. Herein, the fluctuation of the stress–strain curve in Figure 6a can be attributed to local cracks in mechanically weak regions [6]. On the other hand, such fracture behavior means that BZPN foams did not lose integrity after local failure.

### 3.5. Thermal Stability

The images of BZPN foam before/after flame tests are first provided in Figure 7a–d. Even though the flame temperature is over 380 °C, the BZPN foam showed no softening and droplet phenomenon. Moreover, no visible smoke was emitted during the burning test. After the burning test, the glossiness of BZPN foam is reduced, but its macroscopic appearance is well-maintained (Figure 7e,f), indicating its superior flame retardant property.

Comparing the initial SEM image in Figure 5b and those after the burning test in Figure 7g,h, the structure of BZPN foam was warped at different degrees. A small number of holes can be observed in the polymer-rich region, but the main decomposition happened in the polymer-lean region. That is, most of the cells were decomposed, leaving small visible holes on the cell surface. Nevertheless, the cellular outline of BZPN foam can be easily distinguished from Figure 7g,h, indicating the foam structure was not collapsed after the flame test.

Figure 8a,b gives TGA/DTG results of BZPN foams. The temperatures at weight loss 5% (*T_5%_*), 10% (*T_10%_*), maximum decomposition temperature (*T_max_*), and char residue (*C_r_*) at 800 °C, were also summarized in Table 3. In general, with the increasing foaming temperature, BZPN foams showed similar stabilities. This is because the thermal stabilities of the polymer are mainly determined by the polymer host. The decomposition of BZPN foams mainly contains three stages. The first slight weight loss before 200 °C was due to the moisture adsorbed by the porous structure of BZPN foam. Specifically, the existence of TW-80 surfactant in BZPN foam is an amphiphilic compound [39]. The second obvious weight loss is attributed to the degradation of the BZPN macromolecule with a *T_max_* of about 420 °C. The third one is the further carbonation of BZPN foam. However, all BZPN foams exhibited a *T_5%_*, *T_10%_*, *C_r,_* and *T_max_* over 370 °C, 510 °C, 65%, and 420 °C, respectively. It should be mentioned that, in comparison with the previous polymer foams based on polyethylene [14], polystyrene [15], polyimide [16], epoxy [17], and phenolic foams [18], BZPN foams exhibited more outstanding thermal stabilities.

## 4. Conclusions

A novel durable large-size BZPN foam with the exceptional mechanical and thermal properties was realized successfully in the current study, and the main conclusions are as follows:The appropriate foaming temperature of the BZPN foam precursor is below 180 °C, and its viscosity foaming window is below 20 Pa·s.The millet bread-like BZPN foams of decimeter level show uniform spherical isolated cell structure, with the high prospect of large-scale production.BZPN foam involves a hierarchical fracture mechanism during the compressive test and shows a high compression strength over 6 MPa.BZPN foams exhibit superior thermal stabilities with a char higher than 65% at 800 °C, its macroscopic shape was well preserved without the release of visible smoke after/during burning on a lamp.

## Figures and Tables

**Figure 1 polymers-14-05410-f001:**
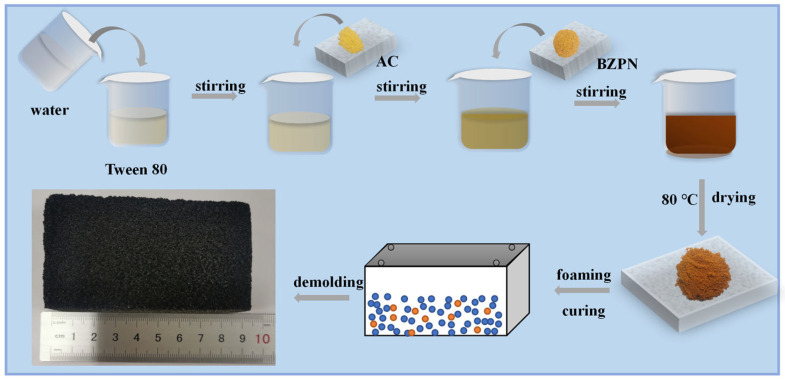
Schematic diagram for preparation process of BZPN foam.

**Figure 2 polymers-14-05410-f002:**
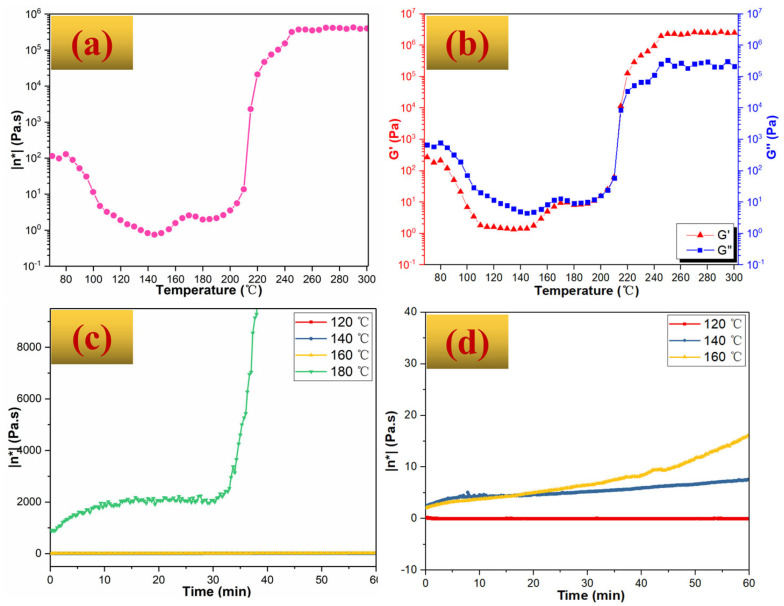
Rheological curves of BZPN/AC/TW foam precursor: (**a**) temperature scan of η*, (**b**) temperature scan of G′ and G″, (**c**) time scan of η*, and (**d**) magnified time scan of η*.

**Figure 3 polymers-14-05410-f003:**
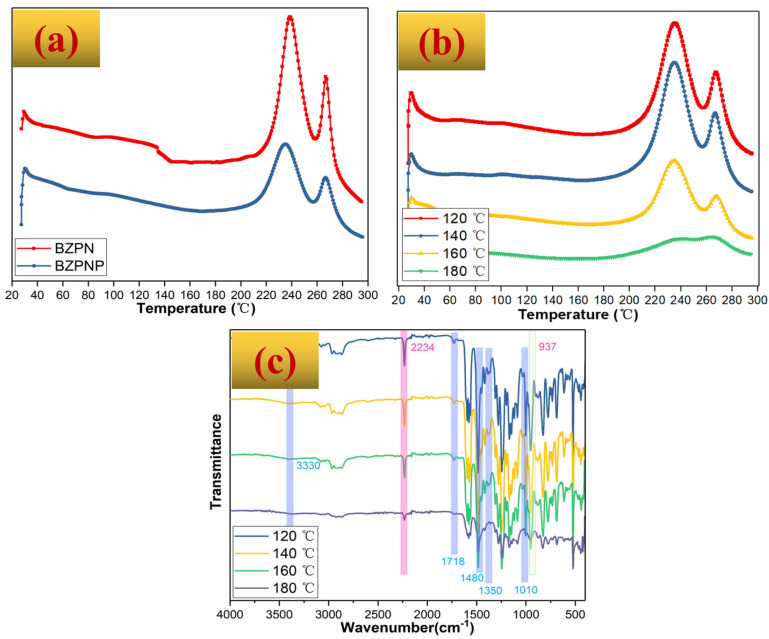
(**a**) DSC curves of BZPN and BZPN/AC/TW foam precursor, (**b**) DSC curves, and (**c**) FTIR curves of BZPN foam at various foaming temperatures.

**Figure 4 polymers-14-05410-f004:**
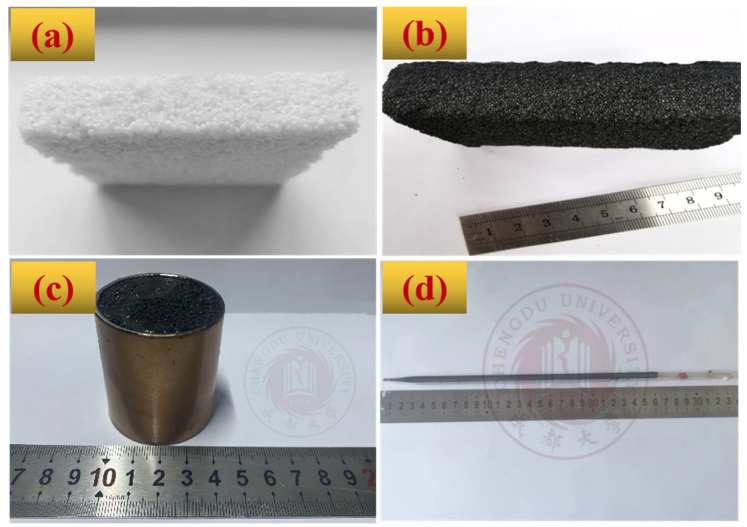
Typical picture of (**a**) millet bread and (**b**–**d**) BZPN foam with different shapes formed at 140 °C.

**Figure 5 polymers-14-05410-f005:**
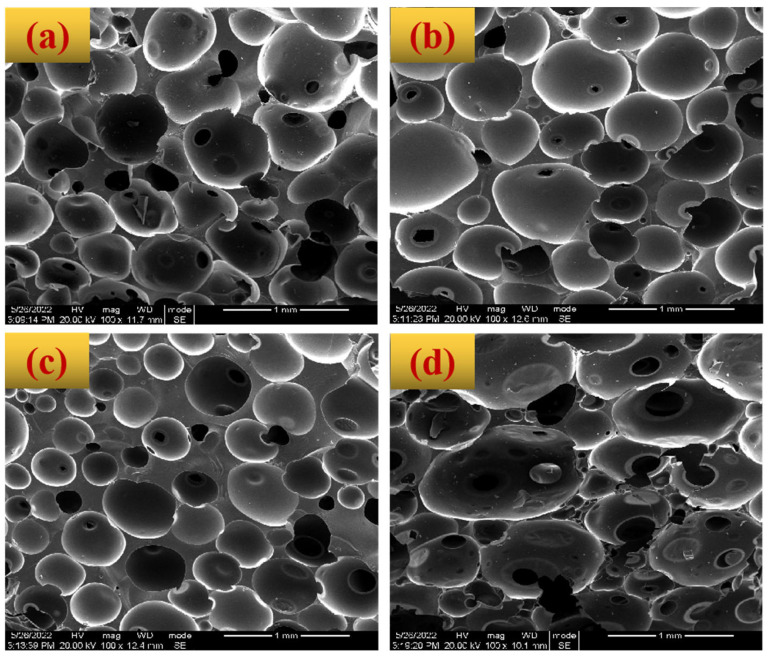
SEM images of BZPN foams formed at various foaming temperature: (**a**) 120 °C, (**b**) 140 °C, (**c**) 160 °C, and (**d**) 180 °C.

**Figure 6 polymers-14-05410-f006:**
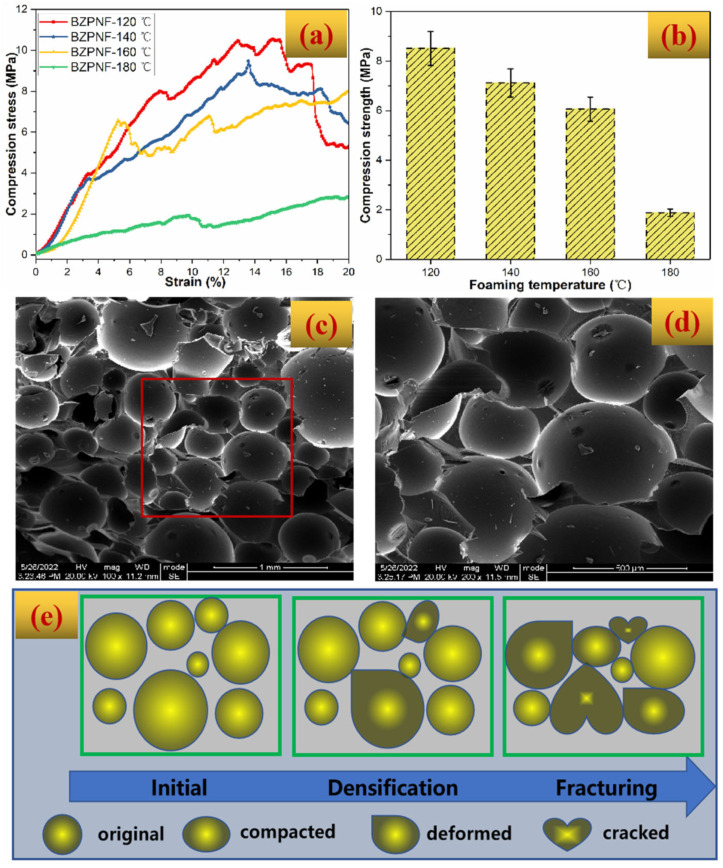
(**a**) Stress–strain curve, (**b**) compressive strength, (**c**,**d**) SEM images after compression test, and and (**e**) fracture mechanism of BZPN foam.

**Figure 7 polymers-14-05410-f007:**
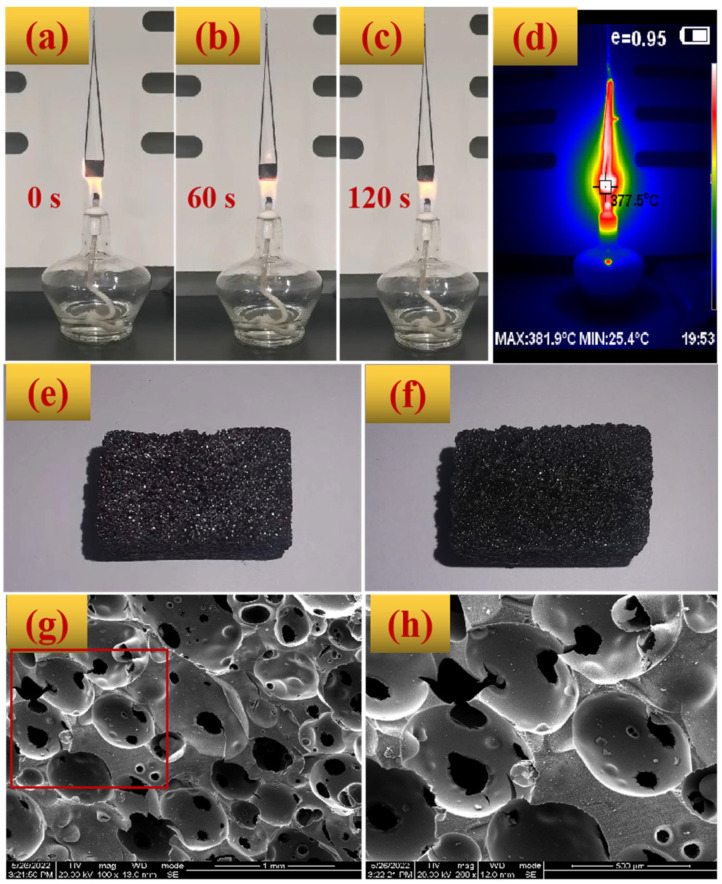
Images of BZPN foams (**a**–**c**) during flame test at 0 s, 60 s, and 120 s, (**d**) real-time infrared image at 60 s, (**e**) image before, and (**f**–**h**) after burning test.

**Figure 8 polymers-14-05410-f008:**
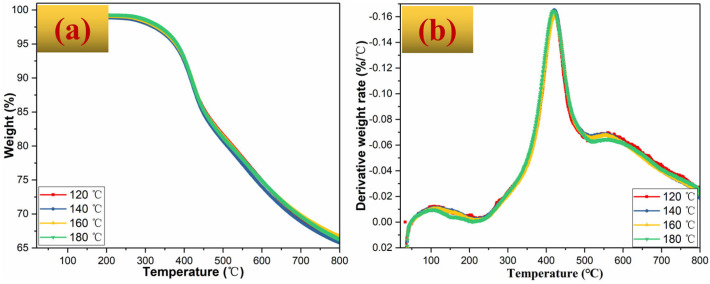
(**a**) TGA and (**b**) DTG curves of BZPN foams.

**Table 1 polymers-14-05410-t001:** Curing procedure of BZPN foams.

Foaming Temperature	Curing Procedure
120 °C	120 °C/2 h, 140 °C/2 h, 160 °C/2 h, 180 °C/2 h, 200 °C/2 h
140 °C	140 °C/2 h, 160 °C/2 h, 180 °C/2 h, 200 °C/2 h
160 °C	160 °C/2 h, 180 °C/2 h, 200 °C/2 h
180 °C	180 °C/2 h, 200 °C/2 h

**Table 2 polymers-14-05410-t002:** Viscosity at a certain time under various temperatures.

Foaming Temperature	Viscosity at a Certain Time (Pa·s)						
	0 min	10 min	20 min	30 min	40 min	50 min	60 min
120 °C	0.041	0.033	0.026	0.033	0.027	0.043	0.052
140 °C	2.547	4.595	4.682	5.218	5.889	6.660	7.548
160 °C	2.103	3.757	4.983	6.470	8.363	11.760	16.180
180 °C	878.2	1949	2067	2068	14,730	54,110	92,280

**Table 3 polymers-14-05410-t003:** Thermal properties of BZPN foams at various foaming temperatures.

Foaming Temperature (°C)	*T_5%_* (°C)	*T_10%_* (°C)	*C_y_* (%)	*T_max_* (°C)
120 °C	376.54	523.67	66.07	422.2
140 °C	374.17	510.51	65.78	422.1
160 °C	380.16	522.44	66.68	421.7
180 °C	380.11	518.33	66.28	421.5

## Data Availability

Data are contained within the article.

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
