# Peer review of "Large-Scale Preparation of Uniform Millet Bread-like Durable Benzoxazine-Phthalonitrile Foam with Outstanding Mechanical and Thermal Properties"

_polymers, 2022, doi:10.3390/polym14245410_

Round 1
Reviewer 1 Report
This paper explains the optimal foaming method of benzoxazine-phthalonitrile foam with better mechanical and thermal properties. This paper has a greater merit in applying the polymer foams at high-temperature applications, and the author performed the studies systematically. I would like to accept the paper with minor corrections to improve the quality of the paper to a further extent.
On page 3 of 14, section number 2.3, the author mentioned about temperature scans at different temperatures, it will be useful for the readers if the ramp rate and soak time are mentioned along with it. In line number 107, in FTIR spectra, the author needs to mention the number of scans since it plays a crucial role in the qualitative representation of FTIR data. In line 115, the author specified it as a cellphone, please do mention the specification of the cellphone camera rather than providing it as just a cellphone. Finally, in line 115 of the same section, the author needs to mention the amount of N2 gas purged throughout the experiment.
On page 7, Fig 3c, The FTIR image is left odd out with out any explanations. The image can be improved to further explain what chemical changes are happening at different temperatures for the BZPN foams.
On page 10, part e, the image needs to be improved, and it doesn't capture the essence of densification in foam. The image of the actual sample at different strain values can be utilized to explain the phenomenon.
Author Response
Dear Reviewers,
Thank you very much for your efforts on our manuscript (polymers-2048080) entitled "Large-scale preparation of uniform milletbread-like durable benzoxazine-phthalonitrile foam with outstanding mechanical and thermal properties". We sincerely thank the excellent reviewers for giving us constructive suggestions that would help us to improve the quality of the paper in depth. With the respect to the reviewer’s comments, our manuscript has been carefully revised and proof-read to minimize problems on grammar, format, etc. as carefully as possible. If there are any other corrections needed, please inform us without any hesitation. Once again thank you and the reviewer for your time and work on our manuscript. Please see the detailed point-to-point response to the Reviewers’ comments in the attached file .
Once again thank you and the reviewer for your endeavors on our work. We look forward to hearing your positive response.
Best wishes,
Yours Sincerely,
Dr. Xulin Yang
yangxulin@cdu.edu.cn
School of Mechanical Engineering,
Chengdu University,
Chengdu, Sichuan 610106, P. R. China.
ORCID link:https://orcid.org/0000-0003-3505-2612

Reviewer 2 Report
The study on the benzoxazine-phthalonitrile foam is interesting and produced good results. However, a few modifications are suggested before the acceptance.
Add the Chemical structure of the Bi-functional gold benzoxazine-phthalonitrile (BZPN) and other raw materials.
The authors did not compare the results with other published data with benzoxazine or phthalonitrile-based foams. like https://doi.org/10.1002/app.51643
A justification is required why the peak at 2234 cm-1 did not disappear entirely even after completing the curing procedure.
why the sample cured at 180 C showed lower compressive strength?
Add bending properties of the foams in the study.
what could be the possible application of the produced foams?
Author Response
Dear Reviewers,
Thank you very much for your efforts on our manuscript (polymers-2048080) entitled "Large-scale preparation of uniform milletbread-like durable benzoxazine-phthalonitrile foam with outstanding mechanical and thermal properties". We sincerely thank the excellent reviewers for giving us constructive suggestions that would help us to improve the quality of the paper in depth. With the respect to the reviewer’s comments, our manuscript has been carefully revised and proof-read to minimize problems on grammar, format, etc. as carefully as possible. If there are any other corrections needed, please inform us without any hesitation. Once again thank you and the reviewer for your time and work on our manuscript. Please see the detailed point-to-point response to the Reviewers’ comments in the attached file.
Once again thank you and the reviewer for your endeavors on our work. We look forward to hearing your positive response.
Best wishes,
Yours Sincerely,
Dr. Xulin Yang
yangxulin@cdu.edu.cn
School of Mechanical Engineering,
Chengdu University,
Chengdu, Sichuan 610106, P. R. China.
ORCID link:https://orcid.org/0000-0003-3505-2612

Round 2
Reviewer 2 Report
Authors incorporated the suggestions, and warrant for publication can be issued.